# Expanding the Phenotype of Hereditary Congenital Facial Paresis Type 3

**DOI:** 10.3390/ijms25010129

**Published:** 2023-12-21

**Authors:** Aysylu Murtazina, Artem Borovikov, Anna Kuchina, Olga Ovsova, Maria Bulakh, Alena Chukhrova, Svetlana Braslavskaya, Oksana Ryzhkova, Nikolay Skryabin, Sergey Kutsev, Elena Dadali

**Affiliations:** 1Research Centre for Medical Genetics, 115478 Moscow, Russia; borovikov33@gmail.com (A.B.); kuchina@med-gen.ru (A.K.); mariya.bulakh@gmail.com (M.B.); achukhrova@yandex.ru (A.C.); braslav_dna@mail.ru (S.B.); ryzhkova@dnalab.ru (O.R.); kutsev@mail.ru (S.K.); genclinic@yandex.ru (E.D.); 2Department of Neurology, Neurosurgery and Medical Genetics, Ural State Medical University, 620028 Ekaterinburg, Russia; ovsovaolga@gmail.com; 3Research Institute of Medical Genetics, Tomsk National Research Medical Center of the Russian Academy of Sciences, 634050 Tomsk, Russia; nikolay.skyabin@medgenetics.ru

**Keywords:** HOXB1, facial paresis, neuromuscular disorder, Moebius syndrome, facio-scapulo-humeral dystrophy

## Abstract

The *HOXB1* gene encodes a homeobox transcription factor pivotal in the development of rhombomere 4. Biallelic pathogenic variants in this gene are associated with congenital facial paresis type 3 (HCFP3). Only seven single nucleotide variants have been reported in the literature to date. Here, we report a 27-year-old female with a unique presentation of HCFP3 with two novel compound-heterozygous missense variants: c.763C>G, p.(Arg255Gly), which arose de novo and an inherited c.781C>T, p.(Arg261Cys) variant. The patient exhibited HCFP3 symptoms with mild upward esodeviation and lacked the documented ear malformations common in HCFP. For many years, she was misdiagnosed with facio-scapulo-humeral muscular dystrophy, due to complaints of shoulder girdle and neck muscle weakness. No alternative genetic or acquired causes of neck and shoulder girdle weakness were found, suggesting its potential inclusion in the phenotypic spectrum.

## 1. Introduction

Currently, three types of hereditary congenital facial paresis (HCFP) have been identified. The first and second types follow an autosomal dominant inheritance pattern with only the mapped loci 3q21-q22 and 10q21.3-q22.1 (MIM 601471 and MIM 604185, respectively). On the other hand, HCFP type 3 (HCFP3) is associated with biallelic variants in the *HOXB1* gene, manifesting at birth with facial muscle weakness and typically accompanied by hearing loss [1]. Additionally, individuals with HCFP3 commonly exhibit ear malformations and some degree of esodeviation [1,2]. Notably, aside from the manifestations related to the 6th, 7th and 8th cranial nerves, no other clinical signs have been reported to date in patients with HCFP3 [3,4,5].

HOXB1 serves as a homeobox transcription factor that plays a key role during embryonic development [1]. It is expressed in the embryonic spinal cord and hindbrain, playing a crucial role in the development of rhombomere 4, which is a precursor of the geniculate ganglion and the spiral and vestibular ganglion [6,7]. To date, seven causative variants have been reported, comprising five missense variants, one frameshift variant, and one nonsense variant [8]. Additionally, there are several cases with different gross deletions encompassing the *HOXB* gene cluster, with more severe and complex phenotypes without clinical features of HCFP3 [9].

Here, we report a Russian patient with novel compound-heterozygous single nucleotide variants (SNVs) in the *HOXB1* gene (NM_002144.4): c.763C>G, p.(Arg255Gly) and c.781C>T, p.(Arg261Cys). Our patient demonstrated a mask-like face due to bilateral facial paralysis, non-progressive and moderate hearing impairment, and a neck muscle weakness that has not been previously reported in patients with HCFP3.

## 2. Results

A 27-year-old woman has been under observation since birth due to congenital weakness of the facial muscles, resulting in a lack of facial expression (Figure 1a). In infancy, she encountered challenges related to sucking and feeding. The proband also reported weakness of the neck muscles, manifesting as a difficulty in maintaining head elevation while in a supine position and sustaining a vertical head position during amusement rides in her early childhood. During her school years, the patient failed to meet the standard requirements for physical exercises, and was unable to perform push-ups and pull-ups. Additionally, she reported a decade-long difficulty with opening bottles and a recent two-year struggle with climbing stairs.

The patient is the first child of healthy, nonconsanguineous Russian parents, born from their second pregnancy. The mother’s first pregnancy resulted in a spontaneous abortion at 3 weeks (Figure 1b). The proband’s birth weight and length were within the normal range. Her motor functions and phrasal speech developed appropriately by the expected age, and there is no recorded history of seizures. From early childhood, a hearing impairment was observed, consistently linked to recurrent episodes of otitis media.

The clinical examination of the patient at the age of 27 years revealed a loss of facial expression due to facial muscles’ paralysis, facial muscle atrophy, lagophthalmos, hypoplasia of the nasal alae, upturned nasal tip, smooth philtrum, downturned corners of the mouth, and prominent nasal voice. She had full eye motility with mild esodeviation when looking up. Salivation and lacrimation were both normal. Additionally, weakness of neck flexors (3/5 MRC) and extensors (4/5 MRC) was noticed. Objectively, no decrease in the strength of limbs or axial muscles was observed, although the patient complained of difficulties in climbing stairs.

All laboratory blood analyses revealed no aberrations. The level of creatine kinase was within normal ranges. Cardiac ultrasound examinations showed normal results. Audiometric assessments demonstrated a second-degree sensorineural hearing loss on both sides.

Comprehensive electrodiagnostic testing was conducted, showing no signs of peripheral neuropathy and any neurogenic or myogenic changes in the limb muscles. Additionally, repetitive nerve stimulation of the facial and axillary nerves displayed no evidence of a neuromuscular junction disorder. However, the blink reflex assessment revealed a bilateral absence of all components, aligning entirely with the characteristics indicative of an impairment in the facial cranial nerve nuclei [10].

The proband underwent brain, neck and limb muscle MRI (Figure 2a–d). Cranial MRI scans revealed no structural anomalies or focal lesions; the vestibulocochlear and accessory nerves were visualized (Figure 2a). At the cervical level, mild hypotrophy of the sternocleidomastoid and trapezius muscles was observed (Figure 2a). However, it is noteworthy that the muscles of the upper and lower limbs appear unaffected on the T1- and T2-weighted images (Figure 2b–d).

The patient had received a muscular dystrophy diagnosis several years earlier due to symptoms including facial muscle weakness, challenges with sucking and feeding, and neck muscle weakness. Furthermore, the patient began expressing concerns about limb weakness at the age of 17, prompting the suggestion of facio-scapulo-humeral muscular dystrophy type 1 as an initial diagnosis. It was excluded using a recently developed PCR-based approach [11], showing the normal range of D4Z4 repeats on both alleles of chromosome 4.

Subsequently, whole exome sequencing (WES) was conducted, revealing two heterozygous missense variants in the *HOXB1* gene: c.763C>G, p.(Arg255Gly) and c.781C>T, p.(Arg261Cys). Given the proximity of these single nucleotide variants (SNVs) to each other, an analysis of the sequence reads demonstrated that both variants were in trans (Figure 3a). Later, Sanger sequencing was performed, confirming the presence of both variants in the proband. The variant c.763C>G was identified in the mother in a heterozygous state, while no variants were detected in the father’s DNA (Figure 3b). As no variants in the *HOXB1* gene were detected in the father, it became imperative to validate paternity within the examined family. Paternity testing was carried out utilizing 16 microsatellite markers. Examination of the samples from the child, her father, and her mother revealed a genetic relationship between the three individuals (Appendix A). This finding supports the hypothesis of a de novo origin of the c.763C>G, p.(Arg255Gly) variant on the father’s chromosome in the proband.

In light of the unexplained neck weakness and complaints of limb muscle weakness, which were not previously linked to variants in the *HOXB1* gene, we pursued additional investigations to ascertain or dismiss the presence of a second underlying condition. Subsequent DNA analysis revealed no expansion of the CCTG repeats in the *CNBP* gene, excluding myotonic dystrophy type 2 as a possibility. Additionally, levels of anti-acetylcholine receptor antibodies and anti-muscle-specific kinase antibodies fell within the normal range. When combined with the normal results of the repetitive nerve stimulation test, these findings allowed for the exclusion of autoimmune myasthenia.

## 3. Discussion

We reported a patient of Russian ancestry diagnosed with a rare form of hereditary congenital facial paresis that expands the clinical spectrum of HCFP3 (Figure 4). To the best of our knowledge, our case presents a first occurrence of variants in a compound heterozygous state within this condition. All previously published cases have been reported with homozygous variants [1,3,4,5,12,13,14]. Through in silico analysis, both novel variants in our patient (p.Arg255Gly and p.Arg261Cys) were projected to substantially impact protein function. Notably, another substitution at the protein residual p.Arg255 was submitted in the ClinVar database. Variant p.Arg255Gln was found in a homozygous state (allele ID 622916) in two siblings with HCFP3, providing additional evidence of its pathogenicity. That record was made with a diagnosis of HCFP3 with intellectual disability, without specifying additional details. In contrast, our proband demonstrates exceptional intellectual development. It is also noteworthy that the p.Arg255Gly variant occurred de novo on the father’s chromosome, which was confirmed by paternity testing.

Furthermore, it is intriguing that all previously reported missense variants in the *HOXB1* gene associated with HCFP3 were substitutions of arginine to other amino acids, as both variants were found in our patient [8]. Both variants are novel, potentially elucidating the presence of additional symptoms previously unobserved in HCFP3 patients. Along with typical phenotypic signs, the patient had a marked weakness of the neck muscles [1]. The complex interplay between *HOXB1* molecular functions and cranial nerve development, especially concerning neck weakness, presents a multifaceted challenge. Interestingly, the occurrence of eye esodeviation, observed in 44% of cases (Figure 4), could not be explained by the function of the 7th and 8th cranial nerves. This prompts speculation that certain mutant isoforms of HOXB1 may influence other cranial nerves. It is plausible that our current understanding of the *HOXB1* gene’s function is incomplete, requiring further fundamental research in this area.

Despite exhibiting the congenital facial paralysis typical of HCFP3, the patient has been diagnosed with muscular dystrophy for several years due to complaints related to weakness in the neck muscles and shoulder girdle. These misdiagnoses induced substantial emotional distress, particularly impacting concerns with regard to pregnancy planning. However, the anxiety was subsequently alleviated following the accurate molecular diagnosis achieved through WES at the age of 26. The molecular findings clarified the primary clinical features of congenital facial paresis and hearing loss. Other possible causes of neck muscle weakness were excluded by providing additional tests. Comprehensive analysis ruled out other variants in the neuromuscular genes, and the examination of D4Z4 repeats revealed normal levels, further supporting the diagnosis of HCFP3 in the proband. We speculate that neck weakness might be a component of HCFP3, yet we acknowledge that additional data are needed to provide concrete evidence of this novel clinical association.

With the aid of WES, a definitive diagnosis was successfully established, enabling us to provide the patient with comprehensive recommendations and engage in a discussion regarding the prognosis of her condition. Given that HCFP3 is a congenital disorder associated with cranial nerves, we anticipate no progression of the muscle weakness or the emergence of new symptoms. Moreover, the patient faces no contraindications for childbirth, and there is a low risk of transmitting the condition to offspring.

The management of this condition primarily involves a symptomatic and preventive approach, as no specific pathogenetic therapy has been developed thus far. Despite the absence of a targeted therapeutic intervention, the confirmed diagnosis has significantly alleviated the patient’s anxiety, representing a noteworthy outcome from our investigation.

## 4. Materials and Methods

The electrodiagnostic studies included motor and sensory nerve conduction studies, repetitive nerve stimulation tests, a blink study and the needle electromyography of limb muscles (deltoid, vastus lateralis, tibialis anterior) using standard techniques. The electrodiagnostics were performed on a Neuro-MEP-micro (Neurosoft, Ivanovo, Russia).

Blood samples were collected from the proband and unaffected parents. Genomic DNA was extracted using standard methods. Whole-exome sequencing was performed on the proband’s DNA to identify mutations. Target enrichment was performed using Illumina TruSeq^®^ ExomeKit (Illumina, San Diego, CA, USA), and custom oligonucleotides (IDT xGen Exome Research Panel v1.0) included coding regions of over 20,000 protein-coding genes. Paired-end sequencing (2 × 150 b.p.) was carried out on an Illumina NextSeq 500. The coding sequence of the *HOXB1* gene was fully covered, with varying coverage depths ranging from 10× to 80×. The sequencing data were processed using Illumina’s Basespace software (Enrichment 3.1.0). The mapped reads were visualized using the Integrative Genomics Viewer (IGV) software (https://www.igv.org/, accessed on 20 November 2023) (© 2013–2018 Broad Institute, Boston, MA, USA, and the Regents of the University of California, San Diego, CA, USA). Variant filtering was based on their frequency, with variants having a frequency of less than 1% in The Genome Aggregation Database (gnomAD v.2.1.1) and coding region sequence effects such as missense, nonsense, coding indels, and splice sites being considered. The clinical significance of the variants was evaluated using ACMG criteria for variant interpretation [15].

To validate the *HOXB1* gene variants in the proband and parents, Sanger sequencing was performed using the Applied Biosystems 3130 xl Genetic Analyzer (HITACHI, Applied Biosystems Group of The Applera Corporation Japan, Waltham, MA, USA). Custom primers (2F: TGGTGGCTAGGTTCAGTTCAGG, 2R: TGGGCAGAATGGCAATGGAG) were used to amplify the fragment encompassing the candidate variant.

Paternity testing was conducted using the “AmpFlSTR Identifiler Direct PCR Amplification Kit” (Applied Biosystems, LLC, USA) which included 16 human DNA loci, following the manufacturer’s protocol. The amplification products were then separated on a 3130 xl Genetic Analyzer capillary electrophoresis instrument.

## 5. Conclusions

Our study expands the understanding of the HCFP3 phenotype by identifying neck weakness as a possible feature within its spectrum. Given the rarity of this disease, with only seven known SNVs to date, our research contributes significantly to its understanding by introducing two novel variants to the mutation spectrum.

## Figures and Tables

**Figure 1 ijms-25-00129-f001:**
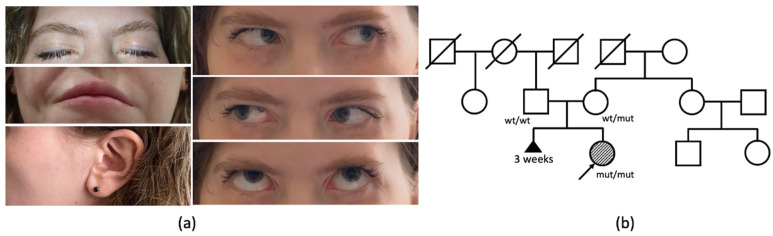
(**a**) Facial appearance of the patient at the age of 27 years. Bilateral lagophthalmos, smooth philtrum, and downturned corners of the mouth are noted, with a normal shape of the auricle and full eye motility with mild esodeviation when looking up; (**b**) pedigree of the reported family.

**Figure 2 ijms-25-00129-f002:**
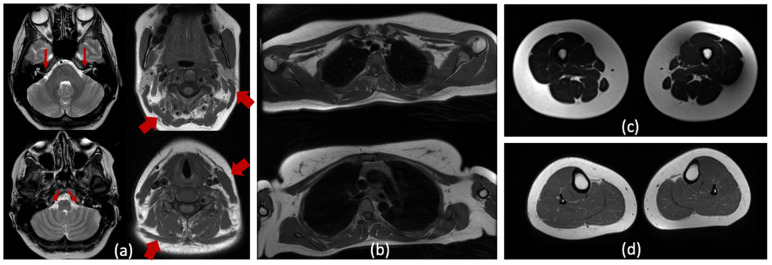
(**a**) Cranial MRI (T2-WI) visualized normal vestibulocochlear and accessory nerves (narrow arrows). Some hypotrophy of sternocleidomastoid and trapezius muscles are noticed on the axial MRI (T1-WI) of the neck muscles (thick arrows); (**b**–**d**) T1-WI (**b**,**d**) and T2-WI (**c**) imaging of the shoulder girdle and leg muscles showed no signs of skeletal muscle involvement.

**Figure 3 ijms-25-00129-f003:**
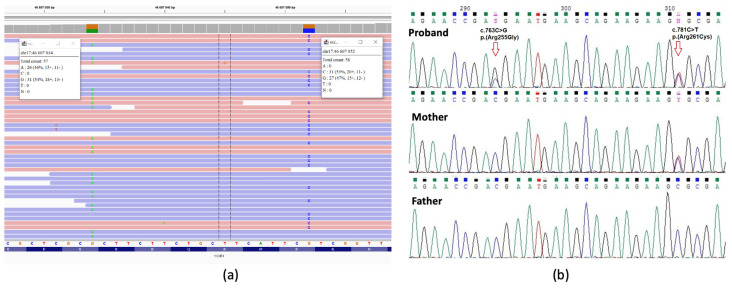
(**a**) Visualization of WES reads in the IGV browser. Both variants in the *HOXB1* gene are located on different sequence reads, suggesting their trans position; (**b**) Sanger sequencing showing both variants, c.763C>G and c.781C>T, in the proband, one variant in the mother and no variants in the father.

**Figure 4 ijms-25-00129-f004:**
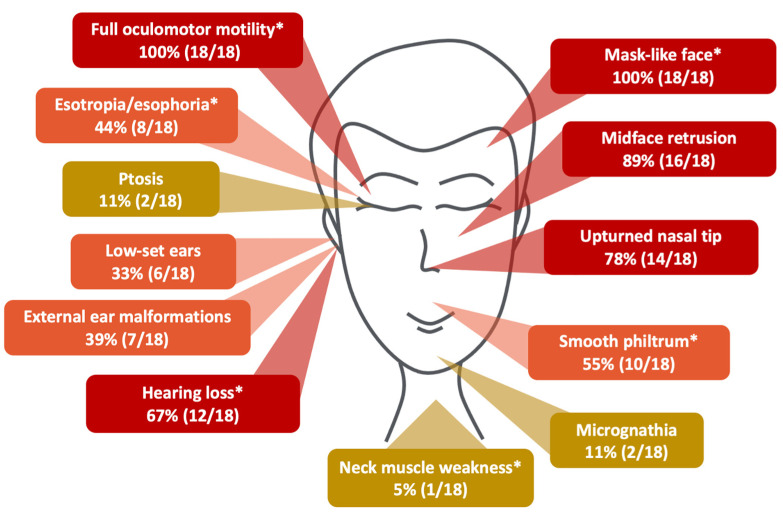
The distribution of HCFP3 signs according to previously and currently reported cases. Red boxes provide the most common features, orange boxes stand for clinical features observed in about half of the patients (30–60% of cases), yellow boxes provide rare signs. The features observed in our patient are marked with an asterisk.

## Data Availability

Data are contained within the article and Appendix A.

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
