# Peer review of "Expanding the Phenotype of Hereditary Congenital Facial Paresis Type 3"

_ijms, 2023, doi:10.3390/ijms25010129_

Round 1

Reviewer 1 Report

Comments and Suggestions for Authors

Here is my view regarding the manuscript:

Strengths:

- Presents a case report of a patient with a novel phenotype of hereditary congenital facial paresis type 3 (HCFP3) caused by novel compound heterozygous variants in the HOXB1 gene. Adds to knowledge of genotype-phenotype correlations in this rare disorder.

- Comprehensive clinical description of the proband including facial features, hearing loss, neck and limb muscle weakness. Includes objective measures like nerve conduction studies, electromyography, MRI imaging. 

- Molecular analysis followed a logical workflow - exome sequencing to identify candidate variants, Sanger sequencing to confirm, and paternity testing to establish de novo occurrence. Appropriate variant interpretation using ACMG criteria.

- Discussion places findings in context of literature on HOXB1 and HCFP3, comparing the phenotypes. Suggests inclusion of neck weakness in spectrum of HCFP3 based on current case.

Weaknesses:

- Sample size of n=1 limits conclusions about neck weakness being part of HCFP3 phenotype. Need more cases to clarify if this is a consistent finding.

- Does not investigate possible mechanism for neck weakness in context of HOXB1 variants. Could speculate on involvement of additional cranial nerves or neuronal populations.  

- Images could be improved for publication quality. For example, Include a scale bar on MRI images, use consistent orientation.

-The results are presented clearly in both the text and figures. The clinical findings are described systematically. The molecular genetic data is visualized effectively to demonstrate the analysis process and confirm the variants. Tables could further organize clinical or genetic data.

Suggested Improvements:  

- Perform additional functional studies of identified variants to demonstrate effects on HOXB1 function. This would strengthen evidence for pathogenicity.

- Screen a cohort of HCFP patients for neck weakness to determine frequency as part of phenotype. Alternatively, review published cases specifically for this symptom. 

- Expand discussion of possible mechanisms underlying neck weakness in context of HOXB1 molecular functions and nerve development.

Overall the manuscript makes a useful contribution towards better definition of the HCFP3 phenotype. Addressing the suggestions above would further improve it for publication. The case highlights importance of molecular diagnosis for proper management of patients with rare diseases.

Comments on the Quality of English Language

 Minor editing of English language required.Some improvements to presentation could enhance clarity.

Author Response

We appreciate the reviewer for taking the time to review this manuscript. Please find the detailed responses below in the attached file and the corresponding revisions/corrections highlighted/in track changes in the re-submitted files.

Reviewer 2 Report

Comments and Suggestions for Authors

This is a case report titled “The Expanding Phenotype of the Hereditary Congenital Facial Paresis Type 3,” focusing on a 27-year-old female patient with a unique presentation of HOXB1-associated disorder. The patient exhibited typical symptoms of Hereditary Congenital Facial Paresis Type 3 (HCFP3), such as mild upward esodeviation and lack of documented ear malformations common in HCFP. She was initially misdiagnosed with facio-scapulo-humeral muscular dystrophy, attributed to complaints of shoulder girdle and neck muscles weakness. However, no alternative genetic or acquired explanations for her neck and shoulder girdle weakness were found, suggesting the need to consider these symptoms in the phenotypic spectrum of HCFP3. The patient’s condition was characterized by compound-heterozygous novel missense variants (de novo c.763C>G p.(Arg255Gly) and inherited c.781C>T p.(Arg261Cys)) in the HOXB1 gene.

The report details the patient’s lifelong struggles with facial muscle weakness, including challenges related to sucking and feeding in infancy, and difficulties in physical activities during her school years. Medical investigations revealed no structural anomalies or lesions in her cranial MRI, normal vestibulocochlear and accessory nerves, and some hypotrophy of neck muscles. Despite specific symptoms of HCFP, her clinical evaluations during life led to a misdiagnosis of muscular dystrophy, causing significant emotional distress. The correct diagnosis was established at age 26 through Whole Exome Sequencing (WES), revealing the HOXB1 gene variants and excluding other neuromuscular disorders. This case contributes to the understanding of HCFP by expanding its phenotypic spectrum to include neck weakness and highlighting the importance of accurate molecular diagnosis in rare genetic disorders.

1. Enhanced Detail on Differential Diagnosis: Provide a more detailed account of the differential diagnosis process, including the reasons for initially considering muscular dystrophy and the factors that eventually led to genetic testing. This would give readers a clearer understanding of the diagnostic challenges and the decision-making process.

2. Broader Literature Review: Include a more comprehensive review of existing literature on HCFP3 and similar disorders. This would help to contextualize the case within the broader field, showing how it contributes new knowledge or confirms existing understandings.

3. In-depth Discussion of Genetic Findings: Expand the discussion around the HOXB1 gene variants, possibly comparing them with other documented cases. This could help in understanding the genetic heterogeneity of the disorder and its phenotypic implications.

4. Patient Management and Follow-Up: Offer more details on the patient’s management plan after the correct diagnosis, including any specific treatments, interventions, or follow-up strategies. This would be beneficial for clinicians encountering similar cases.

5. Impact on Patient and Family: Elaborate on the psychological and social impacts of the misdiagnosis and subsequent correct diagnosis on the patient and her family. This humanizes the case and emphasizes the importance of accurate diagnosis.

Author Response

(The authors gave the same response as above.)
